# Evolution of the Microstructure and Mechanical Performance of As-Sprayed and Annealed Silicon Coating on Melt-Infiltrated Silicon Carbide Composites

**DOI:** 10.3390/ma16124407

**Published:** 2023-06-15

**Authors:** Mengqiu Guo, Yongjing Cui, Changliang Wang, Jian Jiao, Xiaofang Bi, Chunhu Tao

**Affiliations:** 1School of Materials Science and Engineering, Beihang University, Beijing 100191, China; 2AECC Beijing Institute of Aeronautical Materials, Beijing 100095, China; 3Key Laboratory of Advanced Corrosion and Protection for Aviation Materials, Aero Engine Corporation of China, Beijing 100095, China

**Keywords:** melt-infiltrated SiC composite, Si coating, atmospheric plasma spraying, interface, bond strength

## Abstract

In this study, silicon coating was deposited on melt-infiltrated SiC composites using atmospheric plasma spraying and then annealed at 1100 and 1250 °C for 1–10 h to investigate the effect of annealing on the layer. The microstructure and mechanical properties were evaluated using scanning electron microscopy, X-ray diffractometry, transmission electron microscopy, nano-indentation, and bond strength tests. A silicon layer with a homogeneous polycrystalline cubic structure was obtained without phase transition after annealing. After annealing, three features were observed at the interface, namely β-SiC/nano-oxide film/Si, Si-rich SiC/Si, and residual Si/nano-oxide film/Si. The nano-oxide film thickness was ≤100 nm and was well combined with SiC and silicon. Additionally, a good bond was formed between the silicon-rich SiC and silicon layer, resulting in a significant bond strength improvement from 11 to >30 MPa.

## 1. Introduction

Silicon coatings are the most extensively applied coatings as the bond coating of environmental barrier coatings (EBCs), which are specifically designed to protect silicon-based materials, typically ceramic matrix composites (CMCs) consisting of a SiC matrix reinforced with SiC or carbon fibers [1,2]. CMCs are the most promising material to fulfil the role required for the next breakthrough in jet engines [3]. Because CMCs are easily degraded in the presence of water vapor or alkali salts that are common in combustion atmospheres, EBC systems consisting of several layers must be used. The layers include a top coat to prevent reactions with the combustion atmosphere, several intermediate layers to reduce physical and mechanical mismatches, and a bond coat attached to the substrate [4]. Silicon has a similar thermal expansion coefficient (CTE) (4.1 × 10^−6^ °C^−1^) to the SiC substrate (4.7 × 10^−6^ °C^−1^) and can form a highly protective oxide when exposed to wet and dry oxidation conditions at temperatures below 1200 °C. [2] Silicon has been validated as a bonding layer for EBCs based on mullite [5], or mullite plus ceramic compounds such as BaO-SrO-Al_2_O_3_-SiO_2_(BSAS) [6,7], and rare earth monosilicates and/or disilicates [8,9]. Although silicon thin films fabricated using chemical and physical vapor deposition technologies have been intensively studied and widely used because of their excellent properties, atmospheric plasma spraying (APS) is employed for silicon fabrication for economic reasons [10]. APS is the method of spraying molten or heat-softened materials onto a surface to provide a coating. Material in the form of a powder is injected into a very high temperature plasma flame, where it is rapidly heated and accelerated to a high velocity. The hot material impacts on the substrate surface and rapidly cools, forming a coating. It can spray very high-melting-point materials such as refractory metals and ceramics, obtaining a denser and thicker coating.

The performance of the bond coating, especially the bond strength, plays an important role in EBC systems. Although the interlaminar tensile strength of melt-infiltrated SiC composites and the tensile strength of poly-silicon were reported to be 37.5–53.6 MPa [11,12] and more than 1.0 GPa [13,14,15], respectively, the bond between the Si coating and CMC substrate is a challenge yet to be resolved. A Yb_2_SiO_5_/mullite/Si tri-layer EBC was coated on CMC substrates via APS by Yang et al. [16], and the bond strength between the Si layer and CMC was 12.28 MPa. Huang et al. [17] reported a bond strength of about 5.0 MPa because of the failure of CMC substrates. Niu et al. [18] obtained a higher Si bonding of 20.6 MPa using vacuum plasma spraying technology, but a Ti-6Al4V alloy was adopted as the substrate in the study. No higher bond strength of APS Si coating on CMC substrates has been reported. The improvement in process and a full understanding of the bonding between the Si layer and CMC substrate will contribute to the EBC application.

Generally, the bond strength of a coating is related to the residual stress of the coating, its microstructure, and the interface with the substrate, which could be adjusted by heat treatment [19]. Zhang et al. [20] investigated the effects of regular furnace annealing on the residual stress of polysilicon thin film. The results showed that after annealing at a temperature varying from 600 to 1100 °C, the film stress decreased gradually with increasing temperature. According to previous research, the surface tensile stress of the APS Si layers changed into compressive stress after annealing above 1100 °C [21]. Chen et al. [22] reported that isolated-particle healing improved the oxidation resistance of silicon coatings by approximately 24%. Furthermore, the annealing of silicon materials is also employed in some industries [23,24].

In this study, silicon bond coatings were deposited on melt-infiltrated SiC composites using APS and then annealed at 1100 and 1250 °C. The microstructure, interface, and bond strength were evaluated to investigate the effect of the annealing process on the silicon layer.

## 2. Materials and Methods

### 2.1. Coating Preparation

SiC fiber-reinforced SiC composites (BIAM, Beijing, China) prepared using melt-infiltration were used as the substrate. The silicon layer was coated on disks (Ø25.4 mm × 3 mm) and rectangular plates to perform bond strength and other tests, respectively. Before depositing the silicon layer, the SiC-CMC specimens were polished using SiC papers, followed by slight sandblasting with 100 mesh Al_2_O_3_ under a pressure of 0.10 MPa.

Silicon coatings were sprayed on the SiC-CMC surface using a Multicoat© APS system (Oerlikon-Metco. Co., Pfäffikon, Switzerland). The thickness of the silicon layer was approximately 70–100 µm, and the detailed APS process parameters are listed in Table 1.

Commercially available silicon powder (99.9% purity) was used in this study. Figure 1 shows the typical morphologies of the silicon feedstock, which shows that the powder exhibited an irregular crystal shape with a particle size of 45–150 μm. Finally, some of the sprayed silicon coatings were annealed at 1100 and 1250 °C (increase rate: 8 °C/min, time: 1, 5, and 10 h) in a muffle furnace with a normal atmospheric environment.

### 2.2. Analysis Methods

The microstructures of the specimens were characterized using transmission electron microscopy (TEM; Talos™ F200X, Thermo Fisher Scientific, Waltham, MA, USA) and scanning electron microscopy (SEM; FEG-SEM, Zeiss Ultra 55, Oberkochen, Germany) with an energy dispersive spectrometer (EDS; Oxford IE350, Oxford, UK), and 15 kV was used in SEM and EDS testing. The accuracy of the elemental analysis of EDS is about 5 at%. Additionally, the porosity of coatings was calculated using Image J software (Version 1.53e).

X-ray diffractometry (XRD; Smart Lab, Rigaku, Tokyo, Japan) at 10−90° with Cu-Kα radiation (λ = 0.15406 nm) was used to analyze the phases of the coatings. The step length and scanning rate were 0.02 and 8°/min, and the tube voltage and tube current were 40 kV and 150 mA, respectively. Finally, quantification of phase content was carried out using the software Jade 6 after the refinement.

The crystal structure and crystal orientation of the silicon layer were analyzed using field-emission (FE)-SEM (Gemini-300, Zeiss, Oberkochen, Germany) equipped with an electron back-scatter diffraction (EBSD) probe (Oxford C-nano high-resolution probe, Oxford, UK), and ATEX^©^ software (Version 4.06) [25] was used to characterize the results. The cross-sections of the silicon coating for the EBSD analysis were sequentially ground with SiC emery paper (grit size 200–2000) and then polished on a polishing machine using 2.5 μm diamond polishing paste. Finally, the specimen surface was further stress-free polished using vibratory polishing.

Nano-indentation and bond strength tests were performed to analyze the mechanical performance of the silicon coatings. Nano-indentation tests using a Berkovich diamond indenter were conducted on the polished cross-sections (TI950 Tribo-Indenter, Hysitron Corporation, Eden Prairie, MN, USA). Each indentation used a maximum load of 50 mN, an equal load rate of 1.67 mN/s for the loading and unloading cycles, and a 10 s hold time. The indentations were performed on three randomly selected spots on the cross-section of the layer without pores and cracks with an interval >30 μm between each indentation.

The bond strengths of the as-sprayed and annealed coatings were measured according to ASTM C633 using an Instron 5882 testing machine. Disk specimens (Φ25.4 mm × 3 mm) coated on one side were used for the bonding strength tests. Before the test, the samples coated on one side were glued to two 25.4 mm-diameter steel dollies using FM^®^ 1000 adhesive films. The tensile stress perpendicular to the coating plane was uniformly increased at 10 MPa/min until the specimen failed. The bonding strength of the coating was calculated using the recorded fracture load. Three samples were tested for each data point to obtain repeatable results.

## 3. Results and Discussions

### 3.1. Surface Microstructures

Figure 2a–c show the surface microstructure of the as-sprayed Si layer in different magnifications. A typical thermal spray coating morphology with a rough surface was observed. However, some submicronic particles, which were <2 µm, were observed on the surface. Silicon easily oxidizes and the submicronic particles on the surface were mainly composed of silicon oxide [10]. The EDS analysis results (Figure 2d) shows that the mass oxygen content of a common area, such as A in Figure 2c, was approximately 2.88 wt%. The oxygen content of the microparticles (B in Figure 2c), as compared to the common area, was approximately 4.96 wt%, which is >1.5-fold higher.

APS, where silicon powders are heated to a molten or partially molten state by plasma, was used in this study. The molten droplets are accelerated to impinge on the substrate at high velocities, which then solidify and form the coating [26]. Figure 3a–d show typical SEM images of the silicon layer surface after annealing under different conditions. Table 2 lists the oxygen content of marked positions as determined using EDS analyses. No noticeable difference in the morphology was observed after annealing at 1100 °C (as shown in Figure 3a,b), and the submicron particles were still observed on the surface. However, Figure 3c,d clearly show that after annealing at 1250 °C, a thin film appeared on the surface and that the submicronic particles gradually combined with the silicon layer. This combination can be explained by references [27,28], which reported the possibility of SiO_2_/silicon decomposition via the apparent reaction of Si + SiO_2_ → 2SiO when the oxygen pressure at high temperatures is low. However, it is believed that the reoxidation reaction 2SiO + O_2_ → 2SiO_2_ occurred rapidly because of the atmospheric annealing conditions. Generally, surface oxidation becomes more severe with an increasing temperature and annealing duration, as shown in Table 2.

Furthermore, XRD was used to evaluate the changes in the silicon layer phase composition caused by annealing, as shown in Figure 4. The results indicate that the deposited coatings were cubic polysilicon and no phase change occurred during the annealing process at 1100 and 1250 °C. The intensity of the (400), (331), and (422) diffraction peaks increased significantly after annealing, as compared to those of the as-sprayed layer. The phases of the coatings were calculated from the diffraction patterns using Rietveld analyses. The results show that evidence of the orthorhombic-SiO_2_ phase was detected after annealing at 1250 °C for 1 h, and the related orthorhombic-SiO_2_ content was 6 and 9% after 5 and 10 h of annealing, respectively. According to the EDS results, the oxidation of the silicon layer annealed at 1100 °C was slight, and the main product was amorphous silica.

### 3.2. Cross-Sectional Microstructures

Figure 5a,b shows cross-sectional back scattered electron images (BSE-SEM) of the as-sprayed silicon layer. A uniform, complete silicon coating with some randomly distributed pores was obtained on the SiC composites by the APS process, and the porosity was about 5.96%. No apparent defects at the coating/substrate interface were observed. The enlarged image in Figure 5b shows that micro-cracks were observed in the layer, which tended to manifest along the boundary of the particles and is a typical characteristic in plasma sprayed coatings due to their technical characteristics. Additionally, the layer exhibited numerous grains with irregular shapes and sizes ranging from nanoscale to several microns.

One TEM specimen was cut from the inside of the Si layer and prepared via focused ion beam (FIB), which was marked in Figure 5b. Figure 5c,e show a TEM image and SAED pattern of the silicon coating, indicating that the silicon layer observed possessed a polycrystalline structure, which agreed with the XRD results measured on the surface. Two particles (indicated by red arrows) were also observed in the viewing field in Figure 5c, and their sizes were 0.249 and 0.423 µm. EDS patterns of the areas marked as 1, 2, and 3 were obtained, and the mass oxygen content was determined to be 1.35, 1.74, and 49.25 wt%, respectively. Considering the size, shape, and higher oxygen content, these submicron particles were assumed to be those observed on the surface (shown in Figure 2) that were embedded in the coating during the layer-by-layer deposition. High-resolution (HR) TEM studies (shown in Figure 5d) were performed at the selected interface marked by the dashed square in Figure 5c. Both the morphology and SAED pattern show that these silicon oxide particles were amorphous. In contrast, the nearby silicon layer exhibited an excellent cubic structure. The calculated interplanar spacing of (111) was 0.321 nm, slightly larger than the theoretical value of 0.314 nm. However, this confirms a weak combination between the polysilicon and amorphous silica particles, which is disadvantageous for the performance of the coating bond. Furthermore, some twins were observed in the layer, as shown in Figure 5c,e. Figure 5f shows a typical high-resolution TEM image and the corresponding fast Fourier transform (FFT) pattern of a twin, and the interplanar spacing of (111) was calculated to be 0.321 and 0.319 nm. 

In general, the crystal structure may have an influence on the mechanical performance of materials. HRTEM observations in several grains could not clearly reveal the features of the coating; therefore, 43 × 50 µm rectangle regions in the cross-section of the silicon layer (dashed rectangle in Figure 6a) were analyzed using EBSD. The EBSD analyses were performed in 0.08 μm steps. Figure 6b shows an inverse pole figure (IPF) along the *Z*-axis, which revealed the crystallographic orientation and the grain distribution of the as-sprayed silicon layer. The colors red, green, and blue represent the [001], [101], and [111] orientations, respectively. No obvious lamellar texture or preferred orientation was observed in the as-sprayed silicon layer. As the thermal spraying proceeded, the silicon powders were heated to a molten or semi-molten state, which then accelerated to impact the substrate at high speed and were crushed into small particles, followed by quick solidification [10]. The grains did not have the opportunity to grow with a preferred orientation, resulting in the formation of an equiaxed polycrystal. Although some large-size grains were observed, Figure 6c indicates that the sizes of most grains were 0–3 μm, and the average size was 1.351 µm.

Figure 6d shows the distribution of the misorientation angle of the overall scanning area. As reported by Liang et al. [29], a high deviation angle of the adjacent grains occurs easily during the rapid cooling solidification process. The fraction of high-angle grain boundaries (HAGBs; >15°) was higher than that of low-angle grain boundaries (LAGBs; 5–15°), and the maximum misorientation angle reached up to 60°. However, a large number of misorientation angles were <5°, which is related to dislocations. It is reported that crystallographic slip is realized by the movement of dislocations along the slip surface, and that dislocations determine the mechanical properties such as strength, hardness, and others [30,31,32]. To further understand the correlation between dislocations and the mechanical properties, the dislocation density was calculated using the geometrically necessary dislocations (GNDs) and ATEX software (Version 4.06). The GND density was first described by Nye in 1953 [33]. Concepts of GND density and dislocation density were developed further by Bilby, Ashby, and Kroner. Total estimation of the GND density involves measurement of the lattice curvature in three dimensions. With knowledge of all possible dislocation types (slip planes and Burgers vectors), a measure of the required dislocation density that would result in the observed curvatures is obtained [34]. The resolution of the geometrically necessary dislocation from EBSD by ATEX software (Version 4.06) could be found in ref [35]. In this paper, the maximum disorientation angle in the calculation was set at 5°. Figure 6e shows that numerous dislocations were distributed in the as-sprayed silicon layer and that the GND density ranged from 3.4 × 10^13^ to 2.0 × 10^15^ m^−2^. Figure 6f shows that the average GND density in the overall scanning area was 5.686 × 10^14^ m^−2^.

Figure 7a–d shows cross-sectional SEM images of the silicon layer after annealing under different conditions with porosity of 6.97%, 6.68%, 6.15%, and 6.47%. Although the porosity of coatings slightly increased after annealing, no obvious microstructural differences were observed after annealing at 1100 and 1250 °C. Additionally, EDS was used to measure the oxygen contents of the as-sprayed and annealed silicon layer in cross-section rectangles no smaller than 100 × 50 µm, and the results are shown in Table 3. For comparison, the oxygen content of the surface of the silicon layers is also listed in Table 3.

Table 3 shows that the oxygen content of all the surfaces exceeded 10 wt% and increased with a higher annealing temperature at the same annealing time. During the heat treatment, the oxygen content of the surface of silicon layer annealed at 1100 °C increased to approximately 27.94 wt% after 5 h of annealing and then stabilized at 10 hrs. Similarly, the oxygen content saturated at approximately 45 wt% at 1250 °C after 5 h, which is close to the theoretical oxygen content of SiO_2_. The trend of oxygen content in the cross-section of the layers was consistent with that at the surface. However, the maximum oxygen content was no more than 3.5 wt% at 1100 and 1250 °C, suggesting that the APS-deposited silicon layer in this study exhibits good oxidation resistance.

The annealed samples were also evaluated using EBSD, and the representative results of the silicon layer after annealing at 1100 °C/1 h and 1250 °C/1 h are shown in Figure 8. Table 4 summarizes the grain size and GND density. Figure 8a,b show that the crystal structure of the layer after annealing at 1100 and 1250 °C remained that of an equiaxed polycrystal with few large-size grains. Figure 8c,d show that the grains grew to an average size of 1.417 µm at 1100 °C/1 h and 1.535 µm at 1250 °C/1 h. The summarized data in Table 4 also indicates that the grain size increased during the annealing process and that the temperature was more critical than the annealing time.

The relative frequency of the 30–40° misorientation angle in the silicon layer annealed at 1250 °C/1 h increased significantly, as compared to that in the as-sprayed silicon layer, as shown in Figure 8f. However, no obvious difference in the misorientation angle distribution was observed between the as-sprayed layer (Figure 6d) and the layer annealed at 1100 °C/1 h (Figure 8e). However, the average GND density increased by 57.7% and 30.7% after annealing at 1100 °C/1 h and 1250 °C/1 h, as shown in Figure 8g,h, respectively. Subsequently, the average GND density stabilized from 1–10 h, and was approximately 9.0 × 10^14^ m^−2^ in the silicon layer annealed at 1100 °C and 7.4 × 10^14^ m^−2^ for that annealed at 1250 °C. Theoretically, dislocations cluster and arrange themselves into dislocation walls under certain conditions, forming subgrain boundaries. The formation of these subgrain boundaries and their transition into HAGBs absorbs and consumes the stored dislocations, decreasing the dislocation density in the grain region. Therefore, a higher dislocation density contributes to the aggregation of LAGBs and the formation of HAGBs, which in turn conversely reduces the dislocation density within the material [29]. Based on the results of the misorientation angle distribution and GND density, it is possible that the dislocation density in the APS silicon layers increased quickly at the beginning of high-temperature annealing. However, it was partly reduced at 1250 °C because of the formation of HAGBs with misorientation angles of 30–40°.

### 3.3. Microstructures at the Interface

Figure 9a,b show the microstructure at the interface of the as-sprayed silicon layer and CMC substrate, respectively. No obvious defects, such as cracks, were observed. Figure 9b allows the details of the interface to be examined. Except for the SiC fibers, a large amount of irregularly shaped particles (marked as A) with a size ≤1 µm were dispersed in the matrix (marked as B). Furthermore, several strips without any particles (marked by white arrows in Figure 9a) were observed to be perpendicular to the interface in the SiC composites (marked as C in Figure 9b). According to the EDS analysis results of areas A, B, and C, shown in Figure 9c–e, the carbon contents in the particle, matrix, and strip were 52.61 (A), 17.56 (B), and 9.31 at% (C), respectively. Due to the infiltration of molten silicon into the SiC fiber/carbon matrix during the melt-infiltration process, melt-infiltrated-SiC/SiC composites may be composed of different phases, such as crystallized β-SiC with a silicon/carbon near stoichiometric ratio of approximately 0.9, unreacted carbon, silicon-rich SiC, and residual silicon phases [36]. Therefore, it can be preliminarily inferred that the dispersed particles (marked as A) are the crystallized β-SiC (silicon/carbon = 47.39/52.61 = 0.9), the matrix (marked as B) is the silicon-rich SiC phase, and the strip (marked as C) is close to the residual silicon phase.

Figure 9f,g show the microstructure at the interface of the silicon layer and the substrate after annealing at 1100 °C/1 h and 1250 °C/10 h, respectively, which are similar to the other annealing conditions. No obvious change was observed, except for the tighter joint and the discontinuous oxide layer. The enlarged image in the upper-right corner of Figure 9g shows that the maximum thickness of the oxide layer after annealing at 1250 °C/10 h was approximately 0.1 µm.

To better understand the interface of the silicon layer and the melt-infiltrated-CMC during the annealing process, FIB was used to prepare several TEM specimens from the sample annealed at 1250 °C for 5 h, which were then characterized. Figure 10a shows a cross-section bright-field TEM image of the sample, which was cut from the normal position, showing dispersed particles and silicon-rich SiC. The dashed red line displays the interface of the APS silicon layer and the CMC substrate. According to the element distribution in Figure 10b, oxygen was concentrated in two regions at the interface near the dispersed particles (marked by arrow A). The silicon distribution map showed no obvious boundary, especially between the silicon-rich areas and silicon layer, such as the zone marked by arrow B. More details were characterized to understand the interface of the APS silicon layer and SiC composites after annealing at 1250 °C/5 h, as shown in Figure 11, Figure 12 and Figure 13.

Figure 11a,c show cross-sectional bright-field TEM images at the interface near the dispersed particles. Combined with the element map in Figure 11b, it is clear that an approximately 133 nm-thick nano-oxide film occurred. HRTEM was performed on both sides of the oxide layer, as shown in Figure 11d,e. In Figure 11d, FFT analysis confirmed that the dispersed particles were β-SiC structures, and the calculated interplanar spacing of (111) was 0.257 nm. Figure 11e indicates no phase change in the silicon layer next to the nano-oxide film. The interplanar spacing of (220) was 0.192 nm, which is very close to the theoretical value of the cubic-silicon phase. Furthermore, the nano-oxide film was an amorphous oxide of silicon, which appeared to exhibit a good combination with both the β-SiC and silicon layers.

Figure 12a,b show cross-sectional bright-field and HRTEM images at the interface near the silicon-rich SiC area, respectively. The FFT data of region ③ indicates that the silicon-rich SiC was still a β-SiC structure, but the crystal plane spacing changed due to the lower carbon content. It should be noted that a nano-oxide film was not observed, and a good bond was formed between the silicon-rich SiC and the silicon layers. It has been reported that a continuous and dense SiC layer with an obvious twin structure is newly formed at the interface between the silicon and SiC composites during the joining process at 1390–1430 °C [37]. The microstructure at the interface of the silicon/SiC obtained in this study was similar to that reported in reference [37], and a twin structure was also observed, as shown in Figure 12b. In reference [37], the author concludes that the new SiC layer may be generated mainly from the dissolution and reprecipitation of the SiC matrix. It is hypothesized that this process also occurred slowly in this study during annealing, which resulted in improved bonding.

The TEM specimen cut from the strip area (similar to Figure 9b, marked as C) with the lowest carbon content was also prepared and analyzed, as shown in Figure 13. Figure 13a–c show a continuous amorphous nano-oxide film at the interface. The thickness of the film was approximately 10–20 nm, occasionally exceeding 100 nm in the regions with pores. HRTEM analyses were performed on both sides of the oxide layer (shown in Figure 13d,e). The FFT analysis and the calculated interplanar spacing of 0.321 µm (Figure 13d) confirmed a residual silicon phase in the melt-infiltrated SiC composites. The reaction of the residual silicon and APS silicon layer was not observed. Fortunately, the nano-oxide film exhibited a good combination with the bilateral silicon and a low growth rate under the conditions studied in this study.

### 3.4. Mechanical Properties of the Silicon Layer

Nano-indentation tests were performed on the cross-sectional samples to evaluate the average mechanical properties of the as-sprayed and annealed silicon layers. Figure 14a shows the load–displacement curves of the samples annealed at 1100 and 1250 °C for 1 h. The curve of the as-sprayed silicon layer exhibited some non-uniformity, possibly due to local defects introduced during the APS process. However, the load–displacement curves were well matched after annealing. Pop-out signals were observed in all the load–displacement curves, which were previously investigated in reference [38].

The average hardness and Young’s modulus of as-sprayed Si layer were 11.83 and 138.75 GPa, respectively, which were comparable to those reported for APS silicon coatings deposited on SiC substrates (H and E of approximately 11 and 120 GPa, respectively) [39]. The hardness of the annealed layer, as compared to that of the as-sprayed Si layer, increased by 22.6 and 16.7% after annealing at 1100 °C and 1250 °C for 1 h, respectively. The hardness remained relatively stable in the samples annealed for 1–10 h. The relationship between the dislocation density and mechanical properties has been studied in metal materials, and it was found that hardness is proportional to the dislocation density level [30,32]. As mentioned above, the GND density increased after annealing at 1100 and 1250 °C (listed in Table 4); therefore, the hardness increase may be related to the higher dislocation density in the annealed silicon layer. Additionally, the Young’s modulus also increased after annealing. Although the value between the different temperatures was very close (approximately 180–190 GPa), the Young’s modulus of the sample annealed at 1250 °C was slightly higher than that of the sample annealed at 1100 °C. These results are contrary to those of the hardness change, which requires further research.

Pulling-off tests were conducted on the as-sprayed and annealed samples according to ASTM C633 to evaluate the annealing effects on the bonding strength of the APS silicon layers. Figure 15a shows the mean value of the bonding strength. The average bonding strength of the as-sprayed silicon layer was 11.0 MPa, which was lower than that of the annealed samples. The average bonding strength of the samples annealed at 1100 °C increased from 17.9 to 32.4 MPa with the annealing time. In contrast, the average bonding strength rapidly increased to >30 MPa after annealing at 1250 °C for 1 h and showed a slight decrease after 10 h.

Figure 15a shows three typical macroscopic fractures. The fracture position of the as-sprayed samples was located at the interface between the silicon layer and the SiC composites, and almost the entire silicon layer was stripped from the substrate. After annealing at 1100 °C for 1 h, part of the thin silicon layer remained on the substrate. After annealing at 1100 °C for 5 h, the fracture position was located inside the silicon layer. The same macroscopic features were observed in the samples annealed at 1100 °C for 10 h and at 1250 °C for 1, 5, and 10 h. The weak position of the silicon layer changed from the silicon/CMC interface in the original state to the internal layer after annealing. That is, the bonding of the silicon layer and the melt-infiltrated SiC composites was effectively enhanced in this study using heat treatment.

Based on the above observations, a diagram illustrating the typical interfacial structure after annealing was constructed, as shown in Figure 15b. As discussed in Section 3.3, three features were observed at the interface, including β-SiC/nano-oxide film/silicon, silicon-rich SiC/silicon, and residual silicon/nano-oxide film/silicon. It is commonly understood that the oxidation at the interface will weaken the bonding, but the nano-oxide film in this study exhibited a good combination with both the SiC and silicon. Most importantly, a good bond between the silicon-rich SiC and silicon layers, labelled as the “interfacial joint”, was formed during the annealing process. It can be concluded that the “interfacial joint” formed at the interface played a key role in improving the bonding strength.

## 4. Conclusions

Silicon layers were deposited on melt-infiltrated SiC composites using APS, which were then annealed at 1100 and 1250 °C for 1–10 h to investigate the effect of annealing on the layer. The microstructure and mechanical properties were evaluated using SEM, XRD, TEM, nano-indentation, and bond strength tests. The following main conclusions were drawn:

(1) The APS-deposited silicon layer annealed at 1100 and 1250 °C possessed a homogeneous polycrystalline cubic structure without phase transition. The annealed APS-deposited silicon layer exhibited good oxidation resistance and the oxygen content in the cross-section was ≤3.5 wt%, which is significantly lower than that at the surface (45 wt% max).

(2) Three features were observed at the interface between the silicon layer and melt-infiltrated CMC composites after annealing, namely β-SiC/nano-oxide film/silicon, silicon-rich SiC/silicon, and residual silicon/nano-oxide film/silicon. The nano-oxide film thickness was ≤100 nm and exhibited a good combination with SiC and silicon. A good bond was formed between the silicon-rich SiC and silicon layer during the annealing process.

(3) The bond strength of the as-sprayed silicon layer was 11.0 MPa and increased to approximately 30 MPa after annealing at 1100 and 1250 °C. The good bond formed in the interface between the silicon-rich SiC and silicon layers played a key role in improving the bonding strength.

(4) The hardness of the as-sprayed silicon layer was 11.83 GPa and increased by 22.6 and 16.7% after annealing at 1100 and 1250 °C for 1 h, respectively. The hardness then remained relatively stable after annealing for 1–10 h. The hardness may be related to an increase in the dislocation density.

## Figures and Tables

**Figure 1 materials-16-04407-f001:**
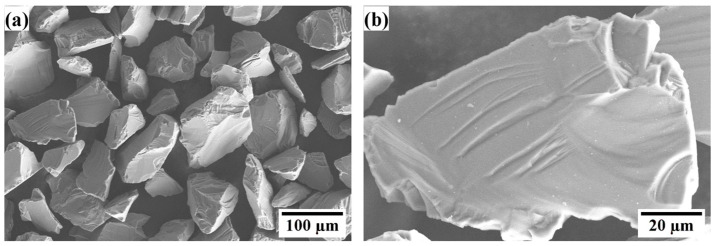
Surface morphology of the silicon powder. (**a**) Secondary electron image of the powder; (**b**) higher magnification image of a single powder particle.

**Figure 2 materials-16-04407-f002:**
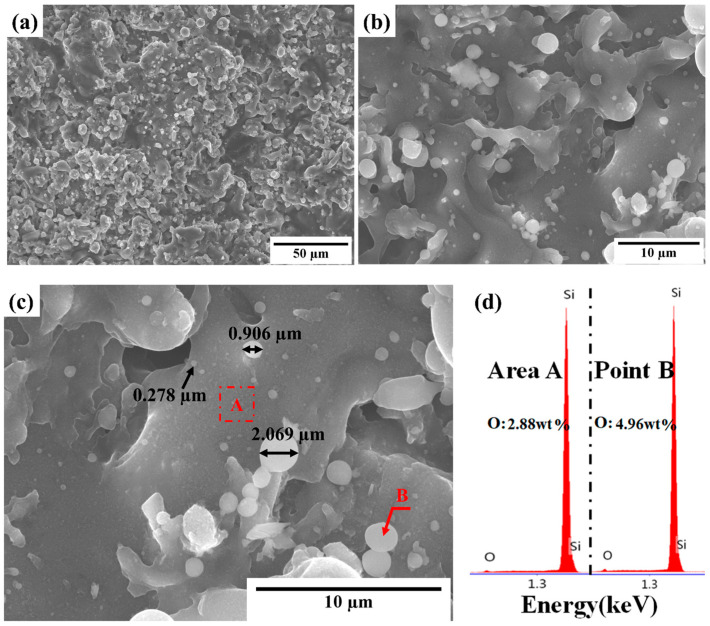
(**a**–**c**) SEM images in different magnification and (**d**) EDS spectra of typical areas of the as-sprayed silicon layer surface.

**Figure 3 materials-16-04407-f003:**
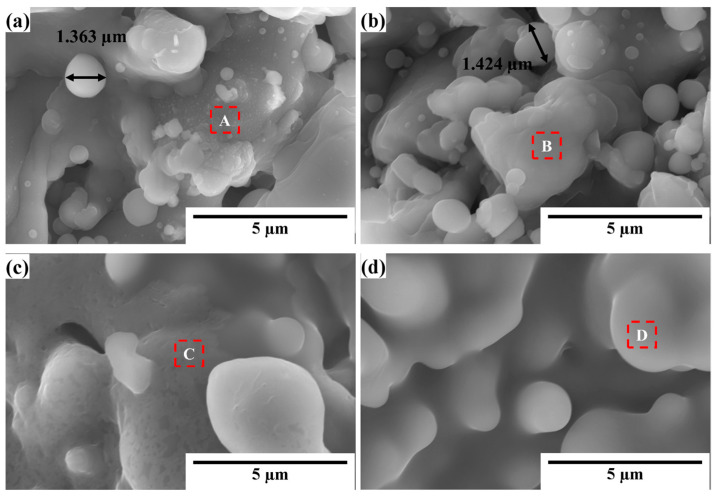
SEM images of the silicon layer surface after annealing at (**a**) 1100 °C/1 h, (**b**) 1100 °C/10 h, (**c**) 1250 °C/1 h, and (**d**) 1250 °C/10 h.

**Figure 4 materials-16-04407-f004:**
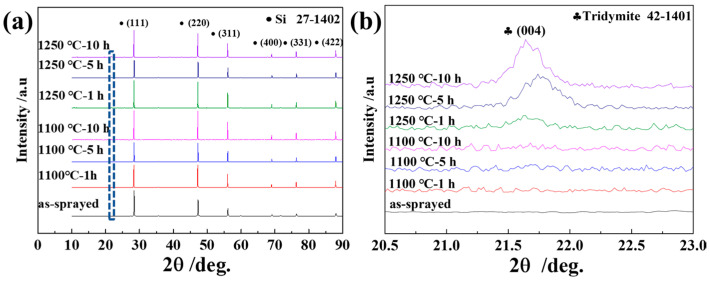
(**a**) XRD measurements of the as-sprayed and annealed silicon layers and (**b**) magnified details at 2θ = 20.5–23.0° (marked by blue frame of (**a**)).

**Figure 5 materials-16-04407-f005:**
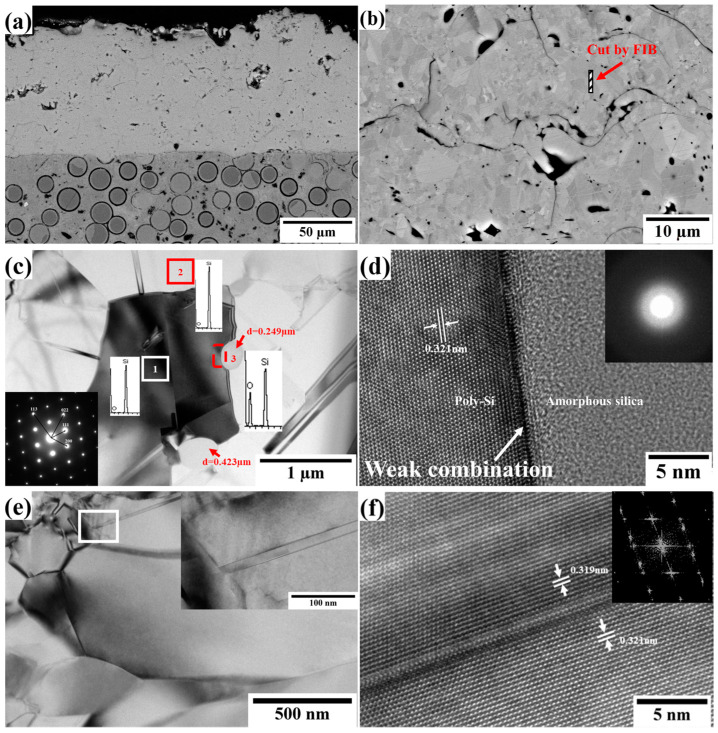
(**a**,**b**) Cross-sectional SEM images and (**c**,**e**) bright-field TEM images of the as-sprayed silicon layers. In (**c**), a selected area electron diffraction (SAED) pattern was obtained in the region indicated by a rectangle marked as 1, and the three EDS patterns shown were obtained in the areas marked as 1, 2, and 3, respectively. The dashed square in (**c**) and solid square in (**e**) are the regions selected for the high-resolution transmission electron microscope (HRTEM) observations shown in (**d**,**f**). The SAED pattern of area 3 in (**c**), enlarged bright-field TEM image of the solid square in (**e**), and the corresponding fast Fourier transform (FFT) pattern of (**f**) are inserted in the upper-right corner of (**d**–**f**), respectively.

**Figure 6 materials-16-04407-f006:**
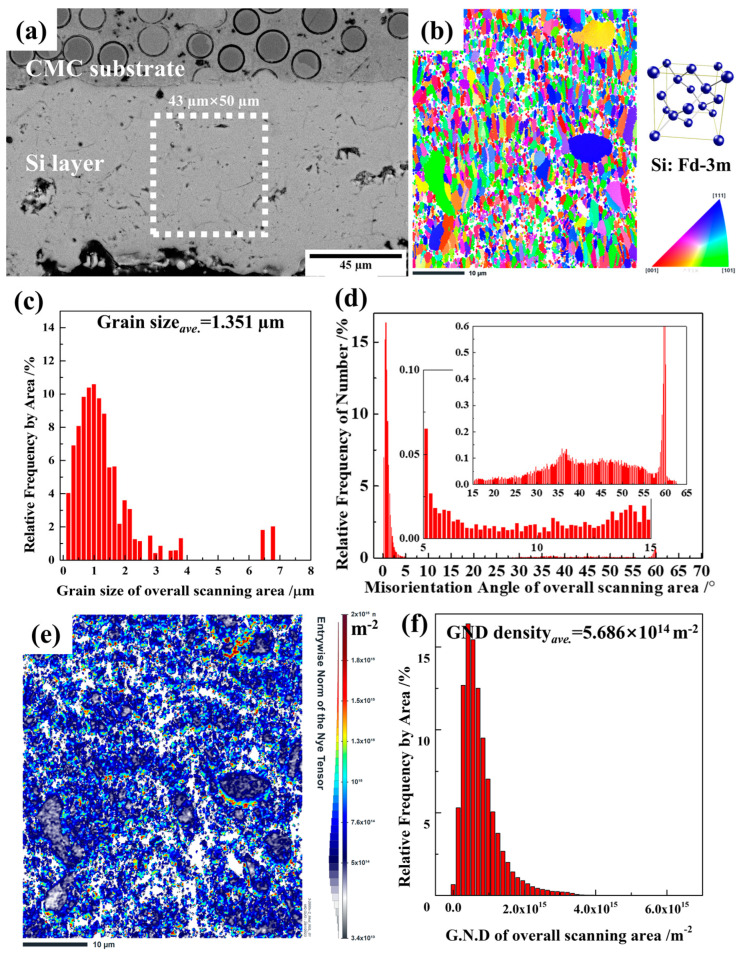
(**a**) The scanning region and results for the as-sprayed silicon layer as determined using EBSD. (**b**) The IPF, (**c**) grain size distribution, (**d**) misorientation angle distribution with two enlarged ranges of 5–15° and >15°, (**e**) GND map, and (**f**) GND distribution.

**Figure 7 materials-16-04407-f007:**
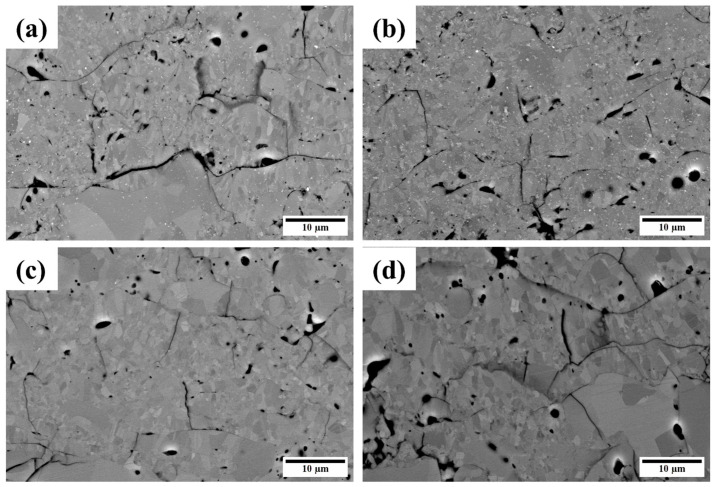
Cross-sectional SEM images of the silicon layers annealed at (**a**) 1100 °C/1 h, (**b**) 1100 °C/10 h, (**c**) 1250 °C/1 h, and (**d**) 1250 °C/10 h.

**Figure 8 materials-16-04407-f008:**
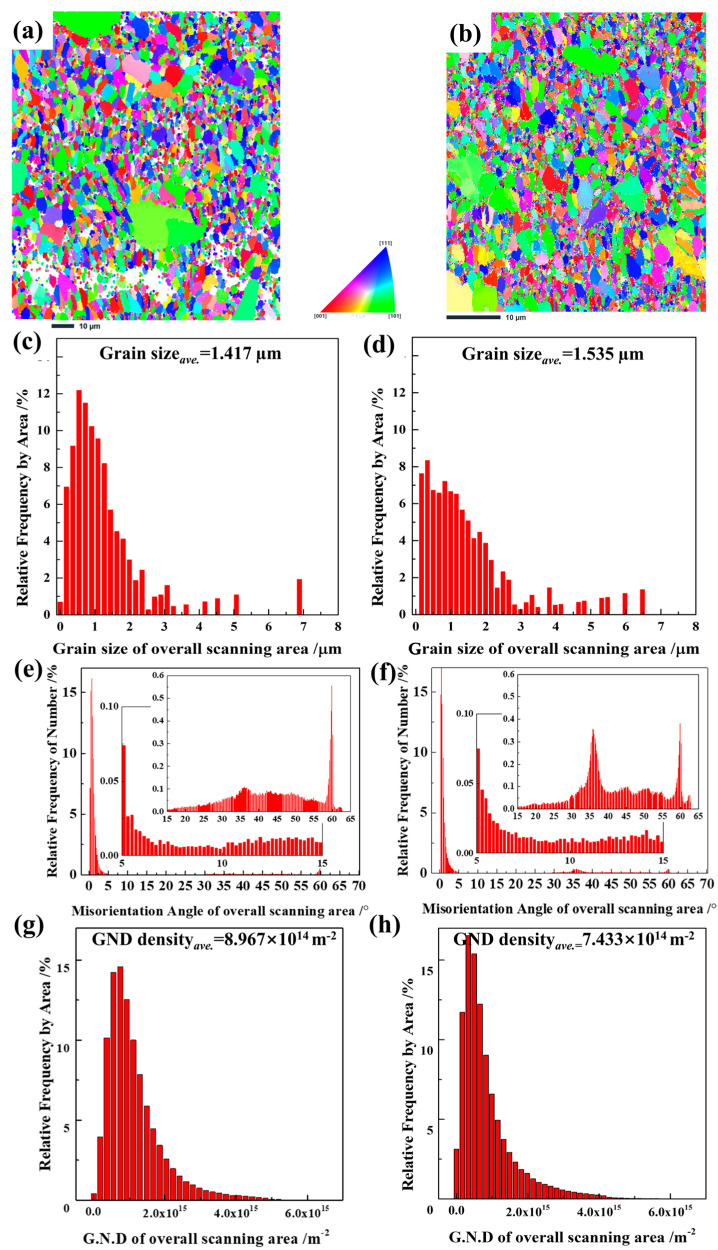
Results for the silicon layer annealed at 1100 °C/1 h (**a**,**c**,**e**,**g**) and 1250 °C/1 h (**b**,**d**,**f**,**h**) as determined using EBSD. (**a**,**b**) IPF, (**c**,**d**) grain size distribution, (**e**,**f**) misorientation angle distribution with two enlarged ranges of 5–15° and >15°, and (**g**,**h**) GND distribution.

**Figure 9 materials-16-04407-f009:**
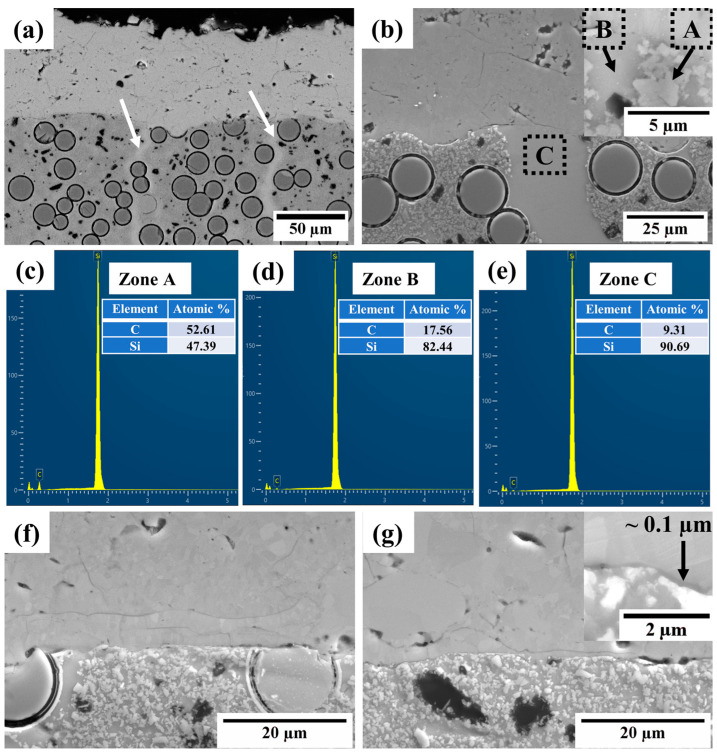
SEM images of the interface between the silicon layer and the CMC substrate. (**a**,**b**) As-sprayed and (**c**–**e**) EDS spectra of zones A–C in (**b**). (**f**) The interface between the silicon layer and the CMC substrate after annealing at 1100 °C/1 h and (**g**) the interface between the silicon layer and the CMC substrate after annealing at 1250 °C/10 h. The insets in (**b**,**g**) show magnified images of the interfaces.

**Figure 10 materials-16-04407-f010:**
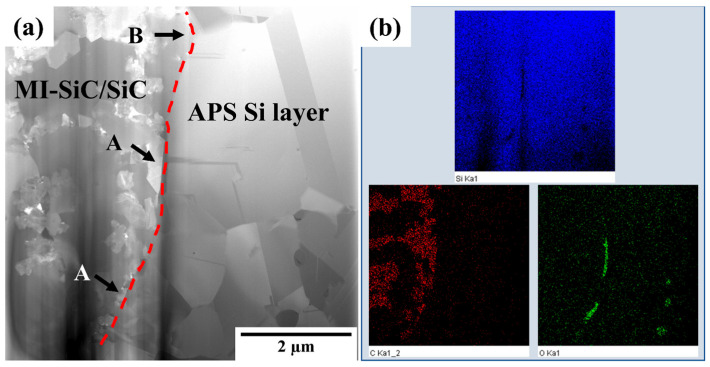
(**a**) Cross-section bright-field TEM image and (**b**) element distribution at the interface after annealing at 1250 °C/5 h as determined using EDS. The crystallized β-SiC particles are marked by arrow A and the silicon-rich SiC phases are marked by arrow B.

**Figure 11 materials-16-04407-f011:**
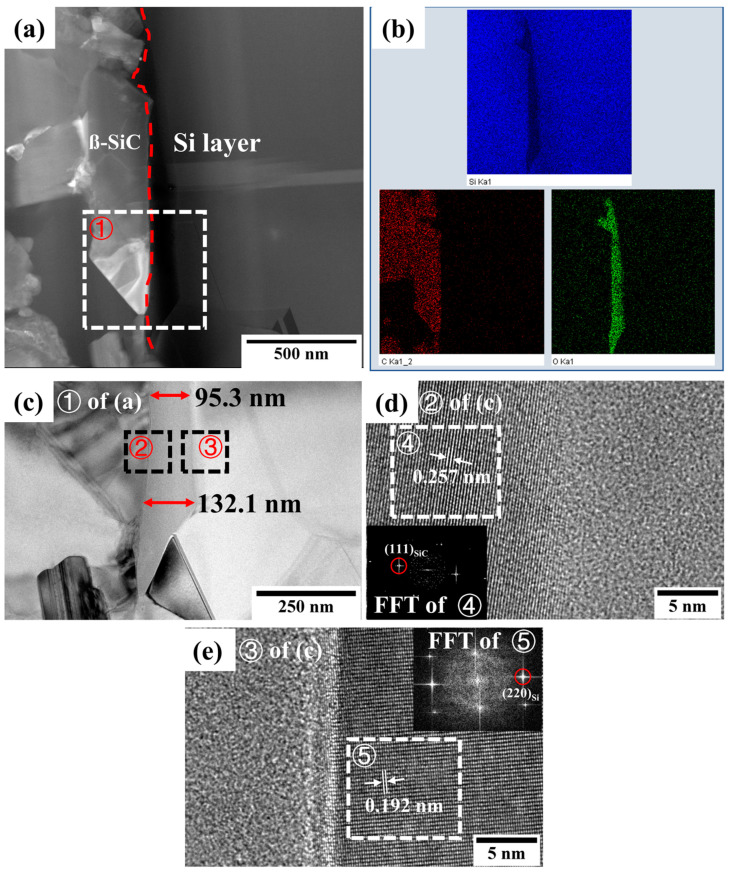
The interface between Si layer and β-SiC after annealing at 1250 °C/5 h. (**a**) Cross-sectional bright-field TEM image at the interface near the dispersed particles. (**b**) Element distribution of (**a**). (**c**) Magnified image of region ①. (**d**) HRTEM image of region ② with the FFT of region ④. (**e**) HRTEM image of region ③ with the FFT of region ⑤.

**Figure 12 materials-16-04407-f012:**
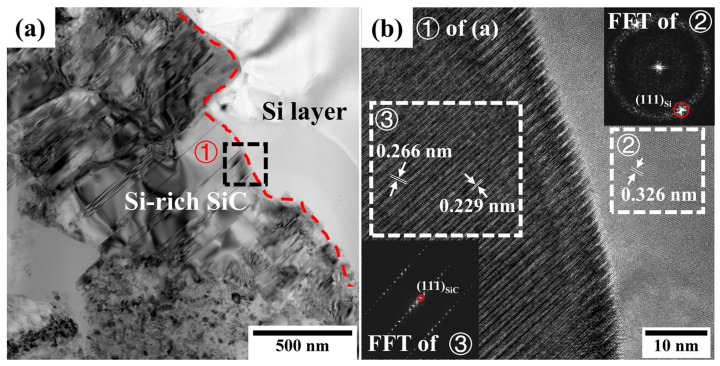
The interface between Si layer and Si-rich SiC after annealing at 1250 °C/5 h. (**a**) Cross-sectional bright-field TEM image at the interface near the silicon-rich SiC and (**b**) HRTEM image of region ① with the FFT of regions ② and ③.

**Figure 13 materials-16-04407-f013:**
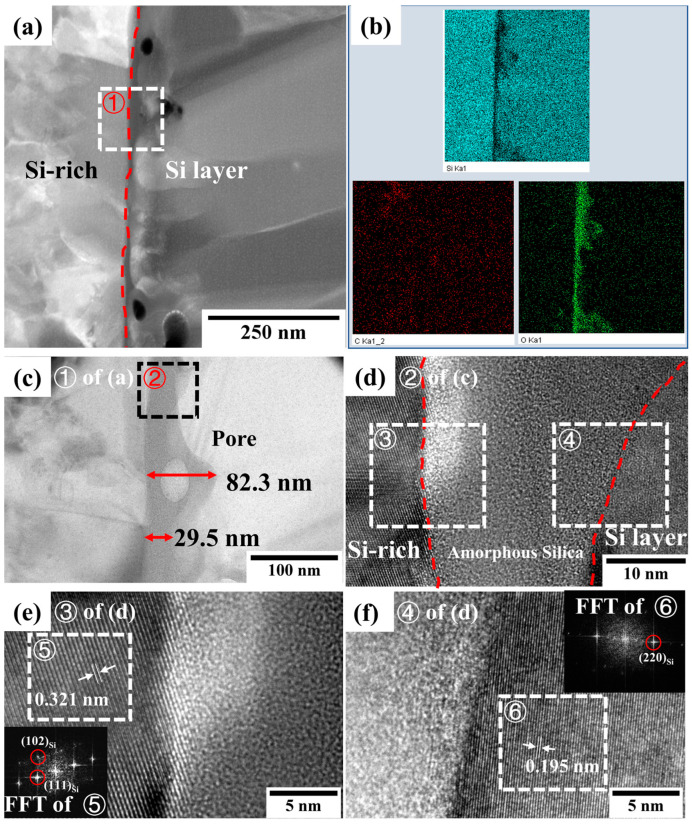
The interface between Si layer and Si-rich after annealing at 1250 °C/5 h. (**a**) Cross-sectional bright-field TEM image at the interface near the strip area (similar to Figure 10b, marked as C) and (**b**) element distribution of (**a**). (**c**) HRTEM image of region ①, (**d**) HRTEM image of region ② with the FFT of region ④, (**e**) HRTEM image of region ③ with the FFT of region ⑤ and (**f**) HRTEM image of region ④ with the FFT of region ⑥.

**Figure 14 materials-16-04407-f014:**
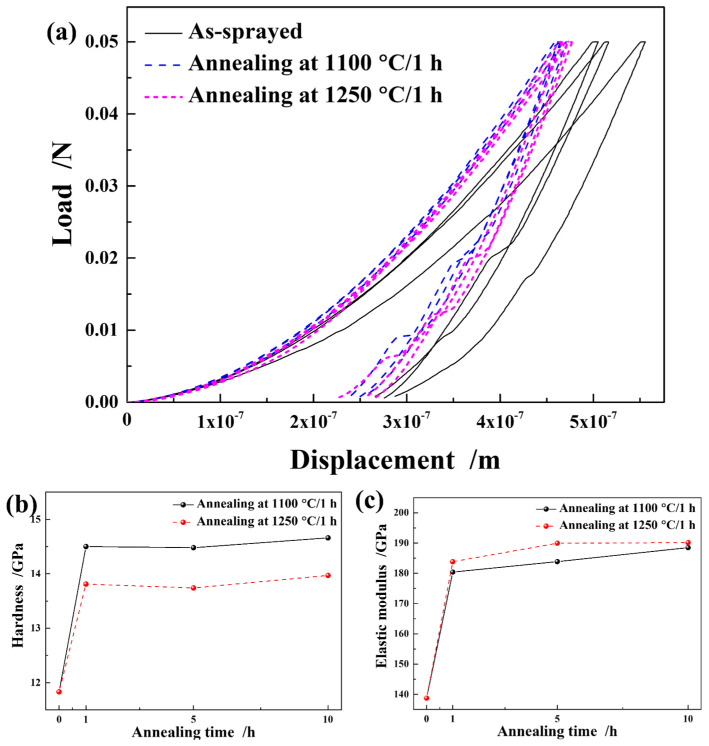
(**a**) Load–displacement curves of the samples under different annealing conditions. The indentation responses of the silicon layers: (**b**) hardness and (**c**) Young’s modulus.

**Figure 15 materials-16-04407-f015:**
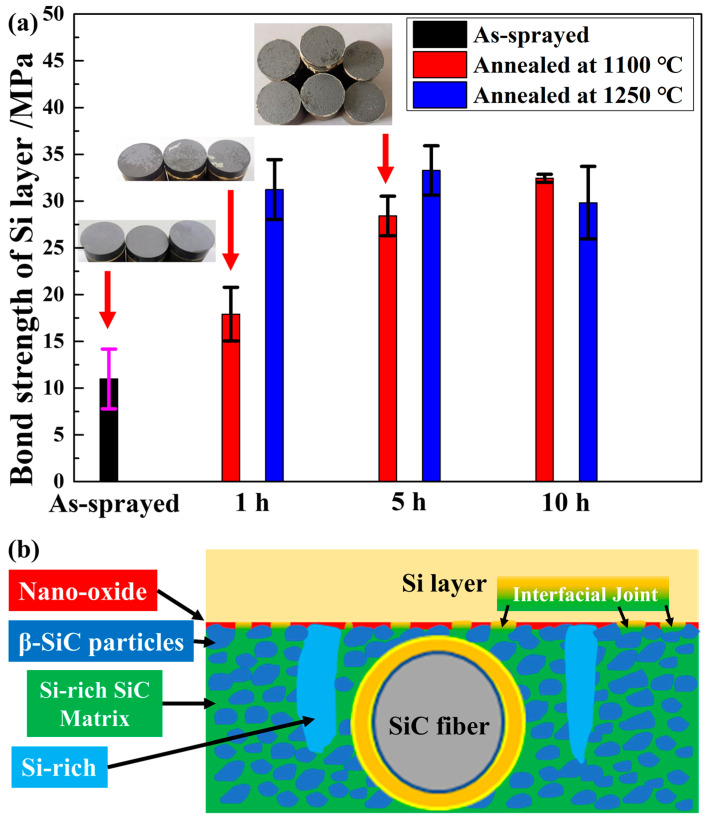
(**a**) Bond strength of the silicon layer under different annealing conditions. Insets show three typical macroscopic fractures. (**b**) Schematic diagram describing the interface evolution between the silicon layer/melt-infiltrated composites after annealing.

**Table 1 materials-16-04407-t001:** APS processing parameters for the silicon coatings.

Items	APS
Current, A	380
Argon, sccm	35,000
Hydrogen, sccm	6000
Feedstock, g/min	25
Distance, m	0.1

**Table 2 materials-16-04407-t002:** The oxygen content at several positions in Figure 3 as determined using EDS.

Position	wt% of Oxygen	Annealing Condition
A	20.11	1100 °C/1 h
B	35.99	1100 °C/10 h
C	20.84	1250 °C/1 h
D	45.01	1250 °C/10 h

**Table 3 materials-16-04407-t003:** The oxygen content of the silicon layers under different conditions as determined using EDS.

wt% of Oxygen on the Surface	wt% of Oxygen in the Cross-Section	Condition
10.08	0.71	As-sprayed
18.60	2.58	1100 °C/1 h
27.94	3.25	1100 °C/5 h
29.11	3.26	1100 °C/10 h
26.95	2.43	1250 °C/1 h
44.69	3.37	1250 °C/5 h
44.23	3.29	1250 °C/10 h

**Table 4 materials-16-04407-t004:** Grain size and GND density of the silicon layers under different conditions as determined using EBSD.

The Average Grain Size (µm)	Average GND Density (m^−2^)	Condition
1.351	5.686 × 10^14^	As sprayed
1.417	8.967 × 10^14^	1100 °C/1 h
1.420	9.011 × 10^14^	1100 °C/5 h
1.424	9.102 × 10^14^	1100 °C/10 h
1.535	7.433 × 10^14^	1250 °C/1 h
1.590	7.381 × 10^14^	1250 °C/5 h
1.611	7.412 × 10^14^	1250 °C/10 h

## Data Availability

The data presented in this study are available on request.

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
