# Peer review of "Evolution of the Microstructure and Mechanical Performance of As-Sprayed and Annealed Silicon Coating on Melt-Infiltrated Silicon Carbide Composites"

_materials, 2023, doi:10.3390/ma16124407_

Round 1

Reviewer 1 Report

The authors present their results on the evolution of microstructure and mechanical properties of atmospheric plasma sprayed silicon coatings, as-sprayed and after annealing of the coatings at high temperatures. A selection of methods was used for microstructural data, followed by an evaluation of the coatings’ mechanical properties via nano-indentation and bond strength tests. Their results leaded to the conclusion of an increased bond strength between the template (melt-infiltrated SiC composite) and the Si layer, with potential application as environmental barrier coating.

While the results are in general well presented, constructing a good scientific case, my opinion is that, at its current form, the paper needs significant modifications before publication. Please find below my suggestions that, I hope, can improve the overall quality of the paper.

Please introduce the acronyms before their first use (see, e.g., BSAS – line 38, EDS – line 92, FIB – line 183, BSE – line 167, SAED – line 194 etc.).

Please make sure that the text within the figures is large enough; for ex., in Figure 4 the text is hardly readable. Similar for Figures 8 c and d. Also, please avoid using low contrast colors (i.e., red on black) – Figures 5c, 11 d (the marking of region 3), 11 e (the marking of region 5), 12 b (the marking of region 3), 13d, e, and f.

Figure 6 is the same as Figure 5. Please replace Figure 6 with the correct one.

Figure 8 – for clarity, please introduce within Figures c-h some info on the sample conditions (e.g., annealed at 1000oC/1 h).

Introduction

Please describe briefly the atmospheric plasma method and its main advantages in comparison with other methods when using it for such coatings (except economic reasons).

Lines 52-53: double negation. I would remove the second “not”.

Subchapter 2.1.

Please explain within the text why two type of samples (disks and plates) were used. Please correlate them with the characterization method they were used for, if the case.

Table 1. Please use Standard International units (e.g., sccm instead of nlpm).

Powder particle size: the range of particle size of the Si feedstock is quite large. Did the authors tried to reduce the range (to get a more uniform distribution of the particle size) by sieving? It should help improving the mechanical strength of the final bond.

Subchapter 2.2.

XRD. Please give more information on the configuration used for the measurements: optics used along the beam (the incidence and receiving components: slits, with or without monochromators, open or closed detector configuration, etc.), parallel beam or Brag Brentano configuration. Slits dimension determines the signal intensity and the width of the diffraction peaks, but also, in case of a close detector configuration (narrow slits), the probability of small intensity peaks to be masked. Was XRD data collected at grazing incidence (as suggested by the used 2 theta axis)? If so, please specify the omega (incidence) angle value.

EDS. Please give information on the accuracy of the elemental analysis (measurements error).

Subchapter 3.1

Lines 139-141. Can the authors explain within the text how they concluded from SEM images that a thin film appeared on the surface?

Line 159. Can the authors explain within the text how they arrived at the mentioned values (6 and 9%) for orthorhombic SiO2 content?

Line 161-162. The authors concluded that “ … the main product was amorphous silica”. However, the XRD data indicate a well define polycrystalline Si layer, with some nanocrystalline SiO2 forming at 1250oC. How the authors concluded that the main product resulted after annealing was amorphous silica?

Subchapter 3.2.

Line 184: “… was marked in Fig. 5(b).” Is this correct? In Fig 5b there is no marking. Please correct.

Line 184: “Figure 5(c) and (e) show a TEM image …” should be replaced with "Figure 5(d) and (f) show a TEM image …”

Lines 184-186: the respective TEM images show part of well-defined Si (111) grains. From these images the authors cannot conclude that the entire silicon layer has a polycrystalline structure, as TEM gives only local crystallographic information. 

Line 192: I would replace “observations” with "studies" or similar.

Line 203: I would remove the word “ generally” after “… crystal structure”. Repetition.

Lines 203-232: the text should be correlated with the correct Figure.

Please give the definition with the text of the “geometrically necessary dislocations” and how it correlates with dislocation density. Please describe briefly how the ATEX software was applied to calculate the dislocation density.

Line 234: A cross-section SEM will give info on films microstructure; the morphology is irrelevant of SEM cross-section images, as it is determined by the specimen preparation for SEM studies. Therefore, my suggestion is to replace “morphological” with “microstructural”.

Line 251: I suggest replacing the word “soaking”, with “annealing”, as soaking has a different meaning.

Line 252: it’s about the oxygen content of the Si layer or of the surface of the annealed Si layer? Please correct.

Lines 258-259: “Additionally, atmospheric ….improves its performances”. Can the authors explain within the text how did they arrive at this conclusion? Otherwise, please remove the phrase.

Table 4 – I would suggest placing it after Figure 8, for better correlation with the text.

Line 267: similar as above for the word “soaking”.

Lines 268-270: please specify within the text the annealing conditions (temperature and time).

Figure 8: within Figure caption and within images, it is unclear which images correspond to the mentioned annealing conditions.

Lines 272-273: The evolution of GND density is also found in Table 4. It should also be mentioned within the text.

Subchapter 3.3.

Line 299. The 9.31 at % of C is not the same value as the one from Figure 9e. Please correct were necessary.

Line 333 – “silicon-rich SiC phase are marked” should read “silicon-rich SiC phase is marked” or “silicon-rich SiC phases are marked”

Formation of the amorphous oxide at the interface. Can this layer be the result of a combination of interface strain and sample preparation for TEM, as it was not formed at the interface between Si-rich SiC and Si film?

Figures 11 and 12: please introduce some info of the specific sample annealing conditions (temperature and time) also in Figure caption.

So, Figure 13 is of an area from as-sprayed sample? Somewhat confusing. Why presenting first TEM on annealed samples and then go back to as-sprayed one?

Subchapter 3.4

Please give a brief description of how the nano-indentation tests were performed.

Line 383: I would replace the word “little” with “some”. The current understanding of the phrase is that there was no non-uniformity in these samples.

Lines 390-392: the phrase refers to as-sprayed samples? Please specify within the text.

Line 414: please replace “soaking” with “annealing”.

Error bars missing in Figure 15 a.

Some concluding remarks.

There are paragraphs of the manuscript with a non-coherent presentation of the results, the authors jumping between studies on as-sprayed and annealed samples. The clarity of presenting the data should be improved in order to make it easier for the reader to follow it.

Oxide layer formation. The authors should discuss, at least briefly, how the oxide layer present at the SiC composite-Si coating interface was formed. Is there any relation between the way the SiC composite disks/plates are prepared before coating and the formation of the oxide layer? How an amorphous layer is formed at the high temperatures used for annealing? Also, how this oxide layer is expected to influence the resulted mechanical properties? It plays the role of a good or of a week bonding point?

There is no correlation between porosity and the evolution of the mechanical properties. Maybe the authors should discuss, at least briefly, what they expect the impact of the porosity will be on the samples mechanical properties.

Good luck!

Kind regards.

Some minor corrections are required.

Author Response

Dear reviewer

Thank you very much for your great work on our manuscript. We have carefully revised the manuscript point by point according to the valuable comments received, and please see the attachment.

The detailed responses to comments are listed as the following:

The authors present their results on the evolution of microstructure and mechanical properties of atmospheric plasma sprayed silicon coatings, as-sprayed and after annealing of the coatings at high temperatures. A selection of methods was used for microstructural data, followed by an evaluation of the coatings’ mechanical properties via nano-indentation and bond strength tests. Their results leaded to the conclusion of an increased bond strength between the template (melt-infiltrated SiC composite) and the Si layer, with potential application as environmental barrier coating.

While the results are in general well presented, constructing a good scientific case, my opinion is that, at its current form, the paper needs significant modifications before publication. Please find below my suggestions that, I hope, can improve the overall quality of the paper.

Point 1: Please introduce the acronyms before their first use (see, e.g., BSAS – line 38, EDS – line 92, FIB – line 183, BSE – line 167, SAED – line 194 etc.).

Response 1: Thanks very much for your careful reading of our manuscript. We introduced the acronyms before their first use. The details are as follows:

  • Line 41: BaO-SrO-Al2O3-SiO2(BSAS)
  • Line 100: Energy Dispersive Spectrometer(EDS; Oxford IE350, UK)
  • Line 180: back scattered electron images (BSE-SEM)
  • Line 192: selected area electron diffraction(SAED)
  • Line 195 High Resolution Transmission Electron Microscope (HRTEM)
  • Line 197: fast Fourier transform(FFT)
  • Line 199-200: Focused Ion beam(FIB)

Point 2: Please make sure that the text within the figures is large enough; for ex., in Figure 4 the text is hardly readable. Similar for Figures 8 c and d. Also, please avoid using low contrast colors (i.e., red on black) – Figures 5c, 11 d (the marking of region 3), 11 e (the marking of region 5), 12 b (the marking of region 3), 13d, e, and f.

Response 2: Thanks very much for your careful reading of our manuscript. We updated some figures. The details are as follows:

  • Line 176: Figures 4 a and b
  • Line 190: Figure 5 c
  • Line 265: Figure 6 d
  • Line 312: Figure 8 e and f
  • Line 389: Figures 11 d and e
  • Line 395: Figure 12 b
  • Line 410: Figure 13 d e and f

Point 3: Figure 6 is the same as Figure 5. Please replace Figure 6 with the correct one.

Response 3: I apologize for the mistakes. Figure 6 should be another figure showing the EBSD result for as-spraying coating, which we have updated.

  • Line 265: Figure 6

Point 4: Figure 8 – for clarity, please introduce within Figures c-h some info on the sample conditions (e.g., annealed at 1000℃/1 h).

Response 4: Thanks very much for your careful reading of our manuscript. We clarified the information on the sample conditions in the Caption of Fig.8.

Line 313:

“Figure 8. Results for the silicon layer annealed at 1100 ℃/1 h(a, c, e, g) and 1250 ℃/1 h(b, d, f, h) as determined using EBSD. (a, b) IPF, (c, d) grain size distribution, (e, f) misorientation angle distribution with two enlarged ranges of 5–15° and > 15°, and (g, h) GND distribution.”

Introduction

Point 5: Please describe briefly the atmospheric plasma method and its main advantages in comparison with other methods when using it for such coatings (except economic reasons).

Response 5: Thanks for your advice. We briefly introduce the method and advantages of the APS in the first paragraph of introduction.

Line 45-50:

“APS is the spraying method of molten or heat-softened materials onto a surface to provide a coating. Materials in form of powder is injected into a very high temperature plasma flame, where it is rapidly heated and accelerated to a high velocity. The hot material impacts on the substrate surface and rapidly cools forming a coating. it can spray very high melting point materials such as refractory metals and ceramics, obtaining a denser and thicker coating. ”

Point 6. Lines 52-53: double negation. I would remove the second “not”.

Response 6: Thanks very much for your careful reading of our manuscript. We removed the second “not”. (Line 61)

Subchapter 2.1.

Point 7: Please explain within the text why two type of samples (disks and plates) were used. Please correlate them with the characterization method they were used for, if the case.

Response 7: Thanks for your advice. The type of samples of disks were used for the bond strength test, and the plates were used for other tests such as X-ray, SEM, TEM etc. we updated the description in Line 80-81.

Line 81-82:

“The silicon layer was coated on disks (Ø25.4 mm × 3 mm) and rectangular plates to perform bond strengths and other tests, respectively.”

Point 8: Table 1. Please use Standard International units (e.g., sccm instead of nlpm).

Response 8: Thanks for your advice. We updated the units in Table. 1: sccm instead of nlpm, and m instead of mm.

Point 9: Powder particle size: the range of particle size of the Si feedstock is quite large. Did the authors tried to reduce the range (to get a more uniform distribution of the particle size) by sieving? It should help improving the mechanical strength of the final bond.

Response 9: Thanks for your advice. As you said, the range of particle size of the Si powder in this work is large. We agree with you that there will be a more interesting performance of Si coating if we reduced the size range. And we are also planning to take this research next. But I am afraid that we have no time to add the next research in this manuscript, because it should be careful planning and implementation.

Subchapter 2.2.

Point 10: XRD. Please give more information on the configuration used for the measurements: optics used along the beam (the incidence and receiving components: slits, with or without monochromators, open or closed detector configuration, etc.), parallel beam or Brag Brentano configuration. Slits dimension determines the signal intensity and the width of the diffraction peaks, but also, in case of a close detector configuration (narrow slits), the probability of small intensity peaks to be masked. Was XRD data collected at grazing incidence (as suggested by the used 2 theta axis)? If so, please specify the omega (incidence) angle value.

Response 10: Thanks for your advice. We have requested the testing laboratory to supplement relevant testing conditions. They used the standard configuration and testing steps of the X-ray diffractometry produced by Rigaku (Model: Smart Lab). Unfortunately, I did not receive more detailed information except the tube voltage and current, which I added in the manuscript.

Line 105-106:

“The step length and scanning rate were 0.02 and 8°/min, and the tube voltage and tube current were 40kV and 150mA, respectively.”

Point 11: EDS. Please give information on the accuracy of the elemental analysis (measurements error).

Response 11: Thanks for your advice. According to the information from the EDS operator, the accuracy of the elemental analysis of EDS is about 5 at%. And we added the data in the manuscript in the first paragraph of Chapter 2.2.

Line 101-102:

“The accuracy of the elemental analysis of EDS is about 5 at%.”

Subchapter 3.1

Point 12: Lines 139-141. Can the authors explain within the text how they concluded from SEM images that a thin film appeared on the surface?

Response 12: Thanks very much for your careful reading of our manuscript. We concluded the thin film based on the area's morphology in Figure 3 (c), similar to the appearance marked C. According to the EDS in Table 2, the O content in the C area is 20.84 wt%, indicating that this is the early stage of surface oxidation of the Si coating.

Figure 3. SEM images of the silicon layer surface after annealing at

 (a) 1100 ℃/1 h, (b) 1100 ℃/10 h, (c) 1250 ℃/1 h, and (d) 1250 ℃/10 h.

Point 13: Line 159. Can the authors explain within the text how they arrived at the mentioned values (6 and 9%) for orthorhombic SiO2 content?

Response 13: Thanks very much for your careful reading of our manuscript. In this part, the contents of orthorhombic-SiO2 were obtained by the X-ray diffraction quantitative analysis using the Jade 6. Also, we added the description in the second paragraph of Chapter 2.2.

Line 107-108:

“At last, quantification of phase content was carried out by the software Jade 6 after the refinement.”

Point 14: Line 161-162. The authors concluded that “ … the main product was amorphous silica”. However, the XRD data indicate a well define polycrystalline Si layer, with some nanocrystalline SiO2 forming at 1250℃. How the authors concluded that the main product resulted after annealing was amorphous silica?

Response 14: Thanks very much for your careful reading of our manuscript. Figure 4(b) shows that tridymite's (400) diffraction peak was obvious after annealing at 1250℃, indicating the crystalline SiO2 formed at 1250℃. On the contrary, no tridymite's (400) diffraction peak was observed at 1100℃. Still, some oxygen element was detected(shown in Table 2), So we concluded that the main product after annealing at 1100 ℃ was amorphous silica.

Subchapter 3.2

Point 15: Line 184: “… was marked in Fig. 5(b).” Is this correct? In Fig 5b, there is no marking. Please correct.

Response 15: Thanks very much for your careful reading of our manuscript. We have added the marking in Fig.5(b).

Point 16: Line 184: “Figure 5(c) and (e) show a TEM image …” should be replaced with "Figure 5(d) and (f) show a TEM image …”

Response 16: Thanks for your advice. We checked figure5 again and confirmed that Fig. 5 (c)(e) were bright-field TEM images, and Fig. 5 (d)(f) were HRTEM images. The text around is the description of Fig.5(c) and (d). I think it is right. Please let me know if you have any more comments on this point.

Point 17: Lines 184-186: the respective TEM images show part of well-defined Si (111) grains. From these images the authors cannot conclude that the entire silicon layer has a polycrystalline structure, as TEM gives only local crystallographic information. 

Response 17: Thanks very much for your careful reading of our manuscript. I think you are right. The “entire” is not rigorous. But combined with the result of XRD(Fig.4 a), TEM(Fig.5 c,e) and SEM(Fig.5 b, numerous grains can be observed obviously), we suppose that most Si coating has a polycrystalline structure. We replaced the “entire silicon layer” with the “silicon layer observed” for more rigour.(Line 201)

Point 18: Line 192: I would replace “observations” with "studies" or similar.

Response 18: Thanks very much for your careful reading of our manuscript. I replace “observations” with "studies" in Line 209.

Point 19: Line 203: I would remove the word “ generally” after “… crystal structure”. Repetition.

Response 19: Thanks very much for your careful reading of our manuscript. We deleted the “generally” in Line 219.

Point 20: Lines 203-232: the text should be correlated with the correct Figure.

Response 20: I apologize for the mistakes. Figure 6 should be another figure showing the EBSD result for as-spraying coating, which we have updated. The text was correlated with the correct Figure 6 now.

Point 21: Please give the definition with the text of the “geometrically necessary dislocations” and how it correlates with dislocation density. Please describe briefly how the ATEX software was applied to calculate the dislocation density.

Response 21: The GND density was first described by Nye in 1953. Concepts of GND density and the dislocation density tensor were developed further by Bilby, Ashby, and Kroner. Total estimation of the GND density involves measurement of the lattice curvature in three dimensions. With knowledge of all possible dislocation types (slip planes and Burgers vectors), a measure of the required dislocation density that would result in the observed curvatures is obtained. In addition, the calculation of GND by ATEX software from the result of EBSD is a complex process, and the details can be found in the reference “Resolving the geometrically necessary dislocation content by conventional electron backscattering diffraction”.

In response to this comment, we have added some descriptions and references about the GND in the manuscript.

Line 244-250:

“The GND density was first described by Nye in 1953[33]. Concepts of GND density and dislocation density were developed further by Bilby, Ashby, and Kroner. Total estimation of the GND density involves measurement of the lattice curvature in three dimensions. With knowledge of all possible dislocation types (slip planes and Burgers vectors), a measure of the required dislocation density that would result in the observed curvatures is obtained.[34] The Resolution the geometrically necessary dislocation from EBSD by ATEX software could be found in ref[35].”

Point 22: Line 234: A cross-section SEM will give info on films microstructure; the morphology is irrelevant of SEM cross-section images, as it is determined by the specimen preparation for SEM studies. Therefore, my suggestion is to replace “morphological” with “microstructural”.

Response 22: Thanks very much for your careful reading of our manuscript. I replace “morphological” with "microstructural" in Line 257.

Point 23: Line 251: I suggest replacing the word “soaking”, with “annealing”, as soaking has a different meaning.

Response 23: Thanks very much for your careful reading of our manuscript. I replace “soaking” with "annealing" in Line 276.

Point 24: Line 252: it’s about the oxygen content of the Si layer or of the surface of the annealed Si layer? Please correct.

Response 24: Thanks very much for your careful reading of our manuscript. It’s about the oxygen content of the surface of the annealed Si layer. I have corrected it in Line 277.

Point 25: Lines 258-259: “Additionally, atmospheric ….improves its performances”. Can the authors explain within the text how did they arrive at this conclusion? Otherwise, please remove the phrase.

Response 25: Thanks very much for your careful reading of our manuscript. This conclusion is inappropriate here. I have removed the phrase.(Line 283-284)

Point 26: Table 4 – I would suggest placing it after Figure 8, for better correlation with the text.

Response 26: Thanks very much for your careful reading of our manuscript. It is my mistake, and I placed Table 4 after Figure 8.

Point 27: Line 267: similar as above for the word “soaking”.

Response 27: Thanks very much for your careful reading of our manuscript. I replace “soaking” with "annealing" in Line 292.

Point 28: Lines 268-270: please specify within the text the annealing conditions (temperature and time).

Response 28: Thanks very much for your careful reading of our manuscript. I have specified the annealing temperature and time within the text in Line 293-294.

Point 29: Figure 8: within Figure caption and within images, it is unclear which images correspond to the mentioned annealing conditions.

Response 29: Thanks very much for your careful reading of our manuscript. We clarified the information on the sample conditions in the Caption of Fig.8.

Line 313:

“Figure 8. Results for the silicon layer annealed at 1100 ℃/1 h(a, c, e, g) and 1250 ℃/1 h(b, d, f, h) as determined using EBSD.

Point 30: Lines 272-273: The evolution of GND density is also found in Table 4. It should also be mentioned within the text.

Response 30: Thanks very much for your careful reading of our manuscript. Like comment point 21, we add some descriptions and references about the GND in the manuscript Line 244-250.

Subchapter 3.3.

Point 31: Line 299. The 9.31 at % of C is not the same value as the one from Figure 9e. Please correct were necessary.

Response 31: Thanks very much for your careful reading of our manuscript. We have corrected Figure 9e.

Point 32: Line 333 – “silicon-rich SiC phase are marked” should read “silicon-rich SiC phase is marked” or “silicon-rich SiC phases are marked”.

Response 32: Thanks very much for your careful reading of our manuscript. We corrected the description to “silicon-rich SiC phases are marked” in Line 364.

Point 33: Formation of the amorphous oxide at the interface. Can this layer be the result of a combination of interface strain and sample preparation for TEM, as it was not formed at the interface between Si-rich SiC and Si film?

Response 33: Thanks very much for your careful reading of our manuscript. At present, there is no evidence or reference to determine that interfacial stress can cause the formation of Si oxides, and the sample preparation for TEM by FIB is in a high vacuum chamber where no oxygen exists. So, we believe the annealing process resulted in the oxidation of Si. Because only less oxygen entered the interface, we think this is the main reason for forming the amorphous oxide instead of orthorhombic SiO2.

Point 34: Figures 11 and 12: please introduce some info of the specific sample annealing conditions (temperature and time) also in Figure caption.

Response 34: Thanks very much for your careful reading of our manuscript. Figures 11, 12 and 13 are also for the sample annealed 1250 ℃/5 h, as same as Figure 10.

We clarified the annealing conditions in the Caption of Figures 11-13. Additionally, we also gave the annealing conditions in the text.

Line 358-360:

“More details were characterized to understand the interface of the APS silicon layer and SiC composites after annealing at 1250 ℃/5 h, as shown in Figs.11–13.”

Point 35: So, Figure 13 is of an area from as-sprayed sample? Somewhat confusing. Why presenting first TEM on annealed samples and then go back to as-sprayed one?

Response 35: Thanks very much for your careful reading of our manuscript. Figure 13 is also for the sample annealed 1250 ℃/5 h, and please see Response 34. Figure 13 shows the third type of interface observed in the sample (residual silicon/nano-oxide film/silicon).

Subchapter 3.4

Point 36: Please give a brief description of how the nano-indentation tests were performed.

Response 36: Thanks very much for your careful reading of our manuscript. We added the test process of nano-indentation briefly in the fourth paragraph of Chapter 2.2.

Line 117-122:

“Nano-indentation tests using a Berkovich diamond indenter were conducted on the polished cross-sections (TI950 Tribo-Indenter, Hysitron Corporation, USA). Each indentation used a maximum load of 50mN, an equal load rate of 1.67mN/s for the loading and unloading cycles, and a 10 s hold time. The indentations were performed on three randomly selected spots on the cross-section of the layer without pores and cracks with an interval > 30 μm between each indentation.”

Point 37: Line 383: I would replace the word “little” with “some”. The current understanding of the phrase is that there was no non-uniformity in these samples.

Response 37: Thanks very much for your careful reading of our manuscript. We replaced the word “little” with “some” in Line 419.

Point 38: Lines 390-392: the phrase refers to as-sprayed samples? Please specify within the text.

Response 38: Thanks very much for your careful reading of our manuscript. Your suggestion is completely accurate. We specified the condition in text.

Line 426:

“The average hardness and Young’s modulus of as-sprayed Si layer were 11.83 and…..”

Point 39: Line 414: please replace “soaking” with “annealing”.

Response 39: Thanks very much for your careful reading of our manuscript. I replace “soaking” with "annealing" in Line 450.

Point 40: Error bars missing in Figure 15 a.

Response 40: Thanks very much for your careful reading of our manuscript. We added the error bars in Figure 15 a.

Some concluding remarks.

Point 41: There are paragraphs of the manuscript with a non-coherent presentation of the results, the authors jumping between studies on as-sprayed and annealed samples. The clarity of presenting the data should be improved in order to make it easier for the reader to follow it.

Response 41: Thanks very much for your comments. We updated some figures and corrected some descriptions according to your valuable comments. I hope the manuscript is easier for the reader now.

Point 42: Oxide layer formation. The authors should discuss, at least briefly, how the oxide layer present at the SiC composite-Si coating interface was formed. Is there any relation between the way the SiC composite disks/plates are prepared before coating and the formation of the oxide layer? How an amorphous layer is formed at the high temperatures used for annealing? Also, how this oxide layer is expected to influence the resulted mechanical properties? It plays the role of a good or of a week bonding point?

Response 42: Thanks very much for your comments. As described in Chapter 2.1, the SiC-CMC specimens were first polished using SiC papers and slightly sandblasted with 100 mesh Al2O3 before depositing the silicon layer, and we don’t think the oxide layer is mainly formed in this process. because the samples were annealed at high temperature in a normal atmospheric environment, both the surface and interface of Si layer oxidized in the annealing process. The main reason for forming the amorphous oxide instead of orthorhombic SiO2 is that only less oxygen entered the interface, and we think the amorphous oxide may transfer to orthorhombic SiO2 if the annealing time is longer. Although the nano-oxide film thickness exhibited a good combination with SiC and silicon, the oxide layer in the interface plays a weak bonding generally. But, during the annealing process, a good bond without oxide layer was formed between the silicon-rich SiC and silicon layer, and we believe that it played a key role in improving the bonding strength.

We described the result briefly in conclusion 3).

Line 496-498:

“The good bond formed in the interface between the silicon-rich SiC and silicon layers played a key role in improving the bonding strength.”

Point 43: There is no correlation between porosity and the evolution of the mechanical properties. Maybe the authors should discuss, at least briefly, what they expect the impact of the porosity will be on the samples mechanical properties.

Response 43: Thanks for your advice. As you said, the pores in the APS Si layers were ignored in the previous work. We calculated the porosity of the coatings using Image J.

Unfortunately, there was no significant trend in the porosity for different heat treatments. But we also listed and compared the porosity in the manuscript. Additionally, we describe the method in chapter 2.2.

The revisions are following:

â‘  At the end of the first paragraph in Chapter 2.2.

“Additionally, the porosity of coatings was calculated by the software of Image J.”

â‘¡ at the first paragraph in Chapter 3.2.

“A uniform, complete silicon coating with some randomly-distributed pores was obtained on the SiC composites by the APS process, and the porosity was about 5.96%.”

â‘¢ at the fifth paragraph in Chapter 3.2.

“Figure 7(a)–(d) shows cross-sectional SEM images of the silicon layer after annealing under different conditions with porosity of 6.97%, 6.68%, 6.15% and 6.47%. Although the porosity of coatings slightly increased after annealing, no obvious microstructural differences were observed after annealing at 1100 and 1250 ℃.”

Comments on the Quality of English Language: Some minor corrections are required.

Response to the comments on the Quality of English Language: Thanks very much for your comment. We have checked the manuscript and revised obvious errors.

I think I have improved my paper, and thanks again for your kind advice.

With best regards,

Mengqiu Guo

On behalf of co-authors

Reviewer 2 Report

This article presents a study on the evolution of microstructure and mechanical performance of silicon coating on SiC composites. The silicon coating was deposited using atmospheric plasma spraying and then annealed at 1100 and 1250 °C for 1 – 10 h. The microstructure and mechanical properties were evaluated using various techniques, including scanning electron microscopy, X-ray diffractometry, transmission electron microscopy, nano-indentation, and bond strength tests. The results showed that a silicon layer with a homogeneous polycrystalline cubic structure was obtained without phase transition after annealing. A good bond between the silicon-rich SiC and silicon layers was formed during the annealing process, which played a key role in improving the bonding strength. The work is well described and everything seems to be measured carefully. I enjoyed a lot reading this work.

I just have minors comments:

a) From the SEM pictures its seems to be many "empty spaces" do the authors measured the mass density of the films and how it was modify by the annealing process? 

I think a fast way to estimate the density is through image analysis and detect the holes as porous. There you could have access to an effective density.

b) Was the annealing done in a special atmosphere? please describe in the main text

c) Figure 5 is the same than 6. It seems that in the text the references to fig 6 are related to fig. 8? Please clarify this in the text and also check the rest of the figures

Along the text there are some strange sentences, typos, subscript not included  

Author Response

Dear reviewer

Thank you very much for your great work on our manuscript. We have carefully revised the manuscript point by point according to the valuable comments received, and please see the attachment.

The detailed responses to comments are listed as the following:

This article presents a study on the evolution of microstructure and mechanical performance of silicon coating on SiC composites. The silicon coating was deposited using atmospheric plasma spraying and then annealed at 1100 and 1250 °C for 1 – 10 h. The microstructure and mechanical properties were evaluated using various techniques, including scanning electron microscopy, X-ray diffractometry, transmission electron microscopy, nano-indentation, and bond strength tests. The results showed that a silicon layer with a homogeneous polycrystalline cubic structure was obtained without phase transition after annealing. A good bond between the silicon-rich SiC and silicon layers was formed during the annealing process, which played a key role in improving the bonding strength. The work is well described and everything seems to be measured carefully. I enjoyed a lot reading this work.

Response: We are pleased that you appreciate our research, and we also hope that this work can provide useful guidance for scholars.

Point 1: From the SEM pictures its seems to be many "empty spaces" do the authors measured the mass density of the films and how it was modify by the annealing process? 

I think a fast way to estimate the density is through image analysis and detect the holes as porous. There you could have access to an effective density.

Response 1: Thanks for your advice. As you said, the pores in the APS Si layers were ignored in the previous work. We calculated the porosity of the coatings using Image J.

Unfortunately, there was no significant trend in the porosity for different heat treatments. But we also listed and compared the porosity in the manuscript. Additionally, we describe the method in chapter 2.2.

The revisions are following:

â‘  At the end of the first paragraph in Chapter 2.2.

“Additionally, the porosity of coatings was calculated by the software of Image J.”

â‘¡ at the first paragraph in Chapter 3.2.

“A uniform, complete silicon coating with some randomly-distributed pores was obtained on the SiC composites by the APS process, and the porosity was about 5.96%.”

â‘¢ at the fifth paragraph in Chapter 3.2.

“Figure 7(a)–(d) shows cross-sectional SEM images of the silicon layer after annealing under different conditions with porosity of 6.97%, 6.68%, 6.15% and 6.47%. Although the porosity of coatings slightly increased after annealing, no obvious microstructural differences were observed after annealing at 1100 and 1250 ℃.”

Point 2: Was the annealing done in a special atmosphere? please describe in the main text.

Response 2: Thanks very much for your careful reading of our manuscript. In this work, the annealing was carried out in the muffle furnace with a normal atmospheric environment. We clarified the description at the end of paragraph 2.1.

The revisions are following:

â‘  At the third paragraph in Chapter 2.2.

“Finally, some of the sprayed silicon coatings were annealed at 1100 and 1250 ℃ (increase rate: 8 ℃/min, time: 1, 5, and 10 h) in a muffle furnace with a normal atmospheric environment.”

Point 3: Figure 5 is the same than 6. It seems that in the text the references to fig 6 are related to fig. 8? Please clarify this in the text and also check the rest of the figures.

Response 3: I apologize for the mistakes. Figure 6 should be another figure showing the EBSD result, which we have updated.

Figure 6. (a) The scanning region and results for the as-sprayed silicon layer as determined using EBSD. (b) The IPF, (c) grain size distribution, (d) misorientation angle distribution with two enlarged ranges of 5–15° and > 15°, (e) GND map, and (f) GND distribution.

Comments on the Quality of English Language: Along the text there are some strange sentences, typos, subscript not included.

Response to the comments on the Quality of English Language: Thanks very much for your comment. We have checked the manuscript and revised obvious errors.

I think I have improved my paper, and thanks again for your kind advice.

With best regards,

Mengqiu Guo

On behalf of co-authors

Reviewer 3 Report

The Article is devoted to the study of silicon coatings obtained by atmospheric plasma spraying. Silicon coating was applied to melt-infiltrated SiC composites and the properties of the coating were studied under various annealing conditions. A number of techniques were used to study coatings, including scanning electron microscopy, X-ray diffraction, transmission electron microscopy, nano-indentation, and bond strength tests. Based on the experiments performed, the Authors draw conclusions about the crystal structure of the coating and the properties of the interface. The article is written quite clearly, but Fig. 6 completely repeats Fig.5. This is a mistake. As a result, the text (lines 218 - 232) remains incomprehensible. This error should be corrected.  The article needs minor revision.

 Minor editing of English language required

Author Response

Response to Reviewer 3 Comments

Dear reviewer

Thank you very much for your great work on our manuscript. We have carefully revised the manuscript point by point according to the valuable comments received, and please see the attachment.

The detailed responses to comments are listed as the following:

The Article is devoted to the study of silicon coatings obtained by atmospheric plasma spraying. Silicon coating was applied to melt-infiltrated SiC composites and the properties of the coating were studied under various annealing conditions. A number of techniques were used to study coatings, including scanning electron microscopy, X-ray diffraction, transmission electron microscopy, nano-indentation, and bond strength tests. Based on the experiments performed, the Authors draw conclusions about the crystal structure of the coating and the properties of the interface.

Point 1: The article is written quite clearly, but Fig. 6 completely repeats Fig.5. This is a mistake. As a result, the text (lines 218 - 232) remains incomprehensible. This error should be corrected.  The article needs minor revision.

Response 1: I apologize for the mistakes. Figure 6 should be another figure showing the EBSD result, which we have updated.

Figure 6. (a) The scanning region and results for the as-sprayed silicon layer as determined using EBSD. (b) The IPF, (c) grain size distribution, (d) misorientation angle distribution with two enlarged ranges of 5–15° and > 15°, (e) GND map, and (f) GND distribution.

Comments on the Quality of English Language: Minor editing of English language required.

Response to the comments on the Quality of English Language: Thanks very much for your comment. We have checked the manuscript and revised obvious errors.

I think I have improved my paper, and thanks again for your kind advice.

With best regards,

Mengqiu Guo

On behalf of co-authors

Round 2

Reviewer 1 Report

The authors corrected the manuscript according to my main suggestions. I believe that the paper can be published in its current form.

Some minor editing still required.